# Identification of drug modifiers for RYR1-related myopathy using a multi-species discovery pipeline

Jonathan R Volpatti[1,2], Yukari Endo[1], Jessica Knox[2,3], Linda Groom[4], Stephanie Brennan[1,2], Ramil Noche[1], William J Zuercher[5], Peter Roy[2,3], Robert T Dirksen[4], James J Dowling[1,2]*

[1]Program for Genetics and Genome Biology, Hospital for Sick Children, Toronto, Canada; [2]Department of Molecular Genetics, University of Toronto, Toronto, Canada; [3]Department of Pharmacology and Toxicology, University of Toronto, Toronto, Canada; [4]Department of Pharmacology, University of Rochester, Rochester, United States; [5]UNC Eshelman School of Pharmacy, SGC Center for Chemical Biology, University of North Carolina, Chapel Hill, United States

**Abstract** Ryanodine receptor type I-related myopathies (RYR1-RMs) are a common group of childhood muscle diseases associated with severe disabilities and early mortality for which there are no available treatments. The goal of this study is to identify new therapeutic targets for RYR1-RMs. To accomplish this, we developed a discovery pipeline using nematode, zebrafish, and mammalian cell models. We first performed large-scale drug screens in *C. elegans* which uncovered 74 hits. Targeted testing in zebrafish yielded positive results for two p38 inhibitors. Using mouse myotubes, we found that either pharmacological inhibition or siRNA silencing of p38 impaired caffeine-induced $Ca^{2+}$ release from wild type cells while promoting intracellular $Ca^{2+}$ release in *Ryr1* knockout cells. Lastly, we demonstrated that p38 inhibition blunts the aberrant temperature-dependent increase in resting $Ca^{2+}$ in myotubes from an RYR1-RM mouse model. This unique platform for RYR1-RM therapy development is potentially applicable to a broad range of neuromuscular disorders.

**\*For correspondence:**
james.dowling@sickkids.ca

**Competing interests:** The authors declare that no competing interests exist.

## Introduction

The ryanodine receptor type I (RyR1) is a calcium release channel located in the terminal cisternae of the sarcoplasmic reticulum (SR) in skeletal muscle. During excitation-contraction coupling (ECC), RyR1 is activated by the voltage sensing L-type calcium channel dihydropyridine receptor (DHPR), located in the transverse tubule (T-tubule) membrane. Together, the T-tubule and two adjacent SR terminal cisternae form a junctional membrane unit referred to as the triad (*Jungbluth, 2007*; *Dowling et al., 2014*; *Jungbluth et al., 2018*). Mutations in the *RYR1* gene are the most common cause of non-dystrophic muscle disease in humans (*Colombo et al., 2015*; *Gonorazky et al., 2018*; *Jungbluth et al., 2018*). *RYR1* mutations are associated with a wide range of clinical phenotypes, collectively referred to as RYR1-related myopathies (RYR1-RM), that can include wheelchair and ventilator dependence, and dynamic symptoms such as exercise induced myalgias, heat stroke, and malignant hyperthermia (*Klein et al., 2012*; *Amburgey et al., 2013*; *Snoeck et al., 2015*; *Jungbluth et al., 2016*; *Matthews et al., 2018*). Despite their relatively high prevalence and associated morbidities, there are currently no approved pharmacological therapies for patients with RYR1-RM.

Much of what is known about the function of RyR1 and the impact of its mutations on skeletal muscle comes from animal models. Well described recessive models of RYR1-RM include the *C.*

**eLife digest** Muscle cells have storage compartments stuffed full of calcium, which they release to trigger a contraction. This process depends on a channel-shaped protein called the ryanodine receptor, or RYR1 for short. When RYR1 is activated, it releases calcium from storage, which floods the muscle cell. Mutations in the gene that codes for RYR1 in humans cause a group of rare diseases called RYR1-related myopathies. The mutations change calcium release in muscle cells, which can make movement difficult, and make it hard for people to breathe. At the moment, RYR1 myopathies have no treatment.

It is possible that repurposing existing drugs could benefit people with RYR1-related myopathies, but trialing treatments takes time. The fastest and cheapest way to test whether compounds might be effective is to try them on very simple animals, like nematode worms. But even though worms and humans share certain genes, treatments that work for worms do not always work for humans. Luckily, it is sometimes possible to test whether compounds might be effective by trying them out on complex mammals, like mice. Unfortunately, these experiments are slow and expensive. A compromise involves testing on animals such as zebrafish. So far, none of these methods has been successful in discovering treatments for RYR1-related myopathies.

To maximize the strengths of each animal model, Volpatti et al. combined them, developing a fast and powerful way to test new drugs. The first step is an automated screening process that trials thousands of chemicals on nematode worms. This takes just two weeks. The second step is to group the best treatments according to their chemical similarities and test them again in zebrafish. This takes a month. The third and final stage is to test promising chemicals from the zebrafish in mouse muscle cells. Of the thousands of compounds tested here, one group of chemicals stood out – treatments that block the activity of a protein called p38. Volpatti et al. found that blocking the p38 protein, either with drugs or by inactivating the gene that codes for it, changed muscle calcium release. This suggests p38 blockers may have potential as a treatment for RYR1-related myopathies in mammals.

Using three types of animal to test new drugs maximizes the benefits of each model. This type of pipeline could identify new treatments, not just for RYR1-related myopathies, but for other diseases that involve genes or proteins that are similar across species. For RYR1-related myopathies specifically, the next step is to test p38 blocking treatments in mice. This could reveal whether the treatments have the potential to improve symptoms.

elegans *unc-68* mutant (null mutant with impaired motility [*Maryon et al., 1996*; *Maryon et al., 1998*]), the *relatively relaxed* zebrafish (loss of function *ryr1b* mutant with impaired motility and early death (*Hirata et al., 2007*), and the 'dyspedic' *Ryr1* null mouse (perinatal lethal [*Buck et al., 1997*; *Avila and Dirksen, 2000*]). In addition, two compound heterozygous mouse models of recessive RYR1-RM were with recently generated and characterized (*Brennan et al., 2019*; *Elbaz et al., 2019*). These models are complimented by 'knock-in' mutants in mice that mirror specific dominant human mutations, including the I4895T mutant (associated with central core disease and referred to as the IT model) (*Zvaritch et al., 2007*; *Zvaritch et al., 2009*; *Lee et al., 2017*), the R163C mutant (associated with malignant hyperthermia) (*Yang et al., 2006*), and the Y522S mutant (associated with malignant hyperthermia and referred to as the YS mouse) (*Chelu et al., 2006*; *Durham et al., 2008*; *Lanner et al., 2012*; *Yarotskyy et al., 2013*).

Previous work using these models identified potential therapeutic targets for RYR1-RM (for a comprehensive review, see *Lawal et al., 2018*) including anti-oxidants (*Durham et al., 2008*; *Dowling et al., 2012*; *Michelucci et al., 2017*), ER stress modulators (*Lee et al., 2017*), and chemicals that influence the binding of RyR1 to modifying partners (e.g. S107, which promotes RyR1/calstabin1 interaction) (*Lehnart et al., 2008*; *Bellinger et al., 2008*; *Andersson et al., 2011*). However, as of yet none of these targets has successfully translated to patients, though *N*-acetylcysteine was tested in a recently completed clinical trial (ClinicalTrials.gov identifier: NCT02362425), where it failed to achieve its primary endpoint (*Todd et al., 2020*). There is thus a critical need to identify and develop new treatment strategies.

With the goal of identifying new therapies for RYR1-RM, we set out to establish a novel multi-species translational pipeline (*Figure 1*). This pipeline is based on the functional conservation of RyR1 across many species, and takes advantage of specific attributes of *C. elegans* (ability to rapidly screen thousands of compounds), zebrafish (large-scale testing in a vertebrate model), and mammalian cell lines (translatability to humans). We screened several thousand compounds, and discovered that p38 inhibition modifies RyR1 phenotypes in all three systems. Our study identifies a new potential therapeutic strategy for RYR1-RM, outlines the utility of multi-species drug discovery, and lays the groundwork for future similar screens for other neuromuscular disorders.

## Results

### Large-scale chemical screen in *C. elegans* identifies 74 unc-68 suppressors

We performed a drug screen using the *unc-68*(*r1162*) *C. elegans* model of RYR1-RM (*Figure 2*). This model has a deletion in the worm ryanodine receptor, lacks RyR protein expression by western blot, and manifests an *unc-68* null phenotype characterized by an 'uncoordinated' (*unc*) movement phenotype, defective pharyngeal pumping, impaired calcium regulation, and reduced fitness (*Maryon et al., 1996*; *Maryon et al., 1998*). We first considered using a liquid-based movement assay (i.e., the *C. elegans* 'thrashing assay' (*Maryon et al., 1996*; *Maryon et al., 1998*) as the basis for our drug screen because *unc-68* mutants thrash at lower rates than WT (*Figure 2—figure supplement 1A*), but found that automatable methods for this assay were sufficiently variable to prevent use in a screen of thousands of compounds. Instead, we developed a sensitized screen based on our observation that nemadipine-A, an inhibitor of the dihydropyridine receptor (*Kwok et al., 2006*),

induces developmental growth arrest in *unc-68* mutants (*Figure 2—figure supplement 1B*). Specifically, *unc-68* worms exposed to 25 μM nemadipine-A arrest at the L1-L3 larval stage, while the majority of wild type N2 strain treated with nemadipine-A have either normal development or, in a small percentage, arrest at the L4 stage (*Figure 2—figure supplement 1B*). Based on this, we screened for chemicals that could overcome this growth arrest by counting the number of L4 and adult stage worms after six days of exposure to both nemadipine-A and chemical (*Figure 2A*).

We evaluated 3700 chemicals in duplicate from a combination of libraries. We screened 770 worm-bioactive (a.k.a. 'wactives') and non-wactives at 7.5 μM and 60 μM, with doses based on bioactivity established in previous screens with *C. elegans* (*Burns et al., 2015*). We also screened 880 kinase inhibitors (GlaxoSmithKline Published Kinase Inhibitor Set) at 60 μM. Lastly, we screened 1280 drugs from the US Drug Collection of clinical trial stage compounds at 60 μM (MicroSource Discovery Systems). The concentration used for the latter two libraries was based on previous *C. elegans* screens performed in the 25–60 μM range (*Kwok et al., 2006*; *Burns et al., 2010*; *Otten et al., 2018*). This concentration range is used to overcome the poor bioaccumulation of exogenous compounds in *C. elegans* (*Burns et al., 2010*).

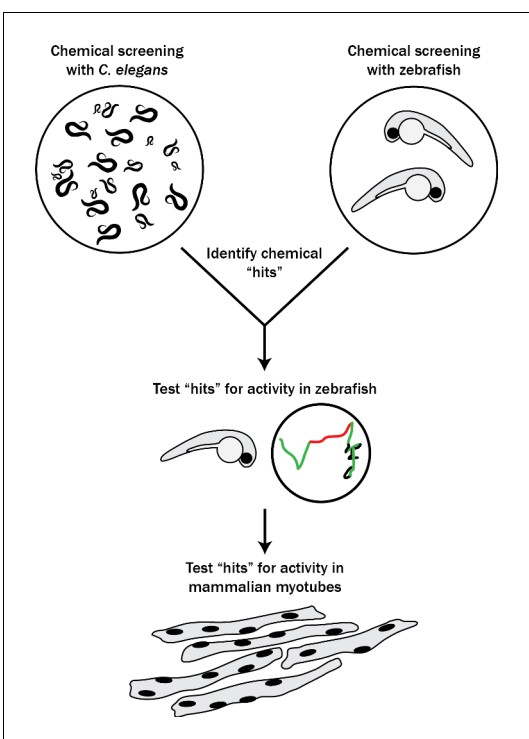

**Figure 1.** Schematic of our multi-species translational pipeline aimed at identifying potential therapeutic targets for RYR1-RM. The pipeline involved screening *C. elegans* and zebrafish with thousands of compounds for suppressors of RYR1 mutant phenotypes, followed by further characterization in zebrafish and evaluation in mammalian cell lines.

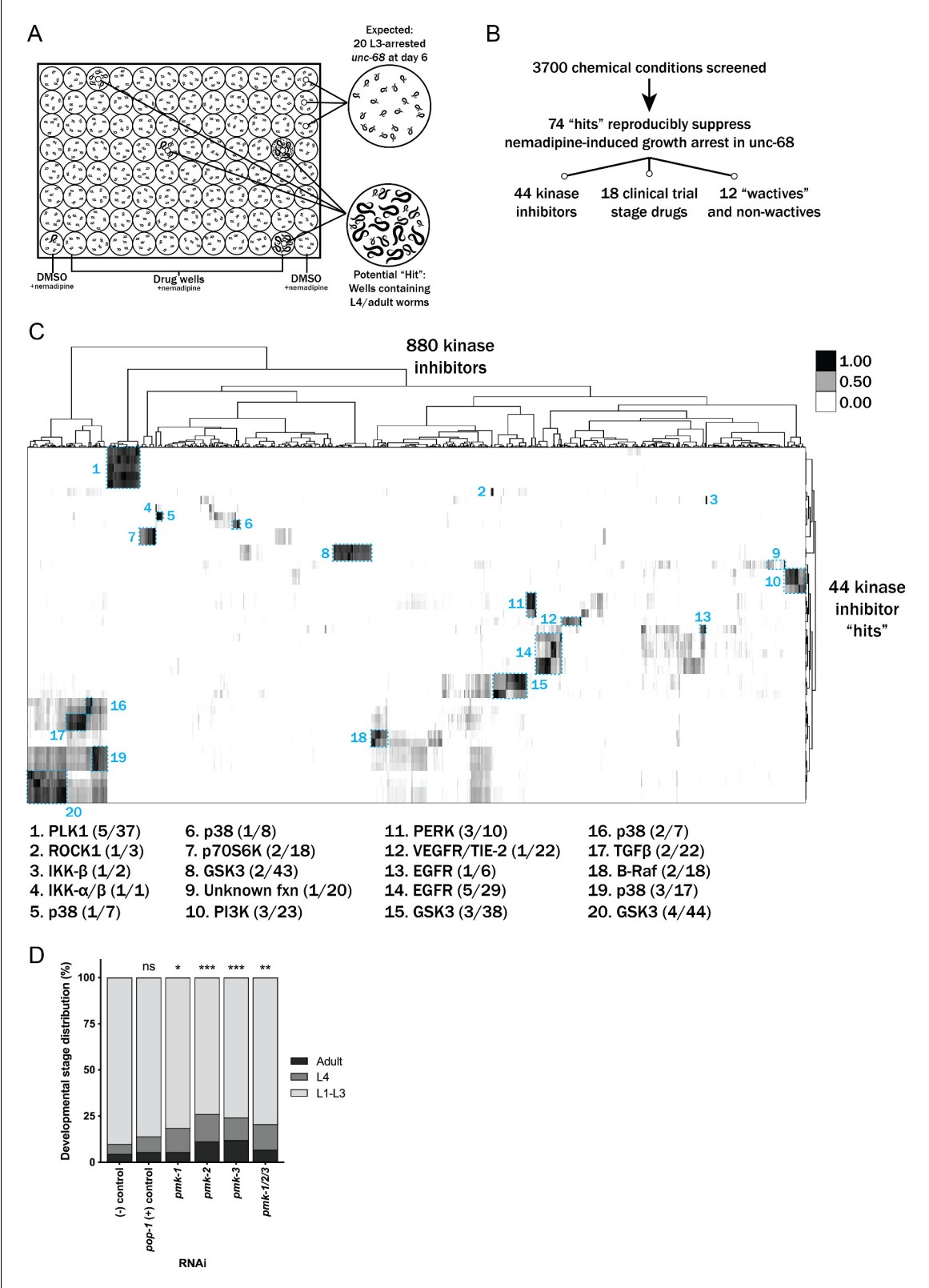

**Figure 2.** Chemical screen finds that p38 MAPK inhibition suppresses the nemadipine-A growth arrest of *unc-68* mutants. (A) Schematic of our screen methodology showing the expected growth arrest phenotype of *unc-68* worms exposed to 25 μM nemadipine after 6 days of exposure and the expected phenotype of a chemical that suppresses this growth arrest. (B) Summary of the 74 'hits' from this screen that reproducibly suppressed nemadipine-induced growth arrest of *unc-68* mutants. (C) Heat map visualization of Tanimoto scores from the 'hit' compounds (y-axis) screened from

*Figure 2 continued on next page*

*Figure 2 continued*

the 880 compounds in the GlaxoSmithKline Published Kinase Inhibitor set (x-axis). Tanimoto scores were calculated for each pair of compounds as a measure of structural similarity and similar clusters were identified via hierarchical clustering of Tanimoto scores (legend indicates the Tanimoto score). As shown, chemicals with similar molecular fingerprints are associated with similar annotated functions/targets. Fisher's exact test was used to determine enrichment based on the number of structurally similar members in each cluster that were either hits or not hits out of the total number of the compounds in the library. (D) RNA interference targeting either *pmk-1, pmk-2, pmk-3*, or a combination of the three shows that p38 MAPK gene knockdown allows a greater proportion of individual *unc-68* worms to escape nemadipine-A-induced growth arrest compared to an empty vector control. Statistical analysis by Kruskal-Wallis test, followed by Dunn's post-test; ns p=0.7360, *p=0.0246, **p=0.0031, ***p<0.001; left-right: *n* = 406, 364, 308, 292, 276, 266.

The online version of this article includes the following figure supplement(s) for figure 2:

**Figure supplement 1.** Phenotypes of *unc-68* mutants considered for chemical screening.

**Figure supplement 2.** Post-hoc statistical analysis showing that 142 chemicals out of the 278 chemicals identified from the primary screen produced developmental stage distributions that were significantly different from random escapee wells (red arrows).

**Figure supplement 3.** Post-hoc statistical analysis showing that 142 chemicals out of the 278 chemicals identified from the primary screen produced developmental stage distributions that were significantly different from random escapee wells (red arrows).

**Figure supplement 4.** Follow-up testing for compounds identified in the primary screen.

**Figure supplement 5.** Heat map visualization of Tanimoto scores from the 'hit' compounds (y-axis) screened from the 1280 compounds in the US Drug Collection library (x-axis).

The primary screen found 278 chemical conditions out of 3700 (~7.5%) that contained at least one L4 or adult worm in duplicate wells (*Figure 2B*). However, 62 single DMSO vehicle control wells out of 760 (~8.2%) and 360 single experimental wells out of 7400 (~4.9%) also contained at least one *unc-68* mutant that reached the L4 or adult stage, which we collectively refer to as 'random escapees'. Based on this, we concluded that many of the 278 chemicals may potentially be false positives. We prioritized 145 of the 278 chemicals for re-testing because they demonstrated in duplicate wells the most complete (i.e. highest number of L4 and adult) suppression the phenotype.

Of note, we only counted the number of 'rescued' worms (L4 and adult) per well, and did not count the exact number of L1-L3 worms in each well. Precise counts would have allowed us to calculate the true proportion of worms which escaped growth suppression and thus more accurately determine which chemicals promoted statistically meaningful growth arrest rescue values as compared to random escapee wells. We instead addressed this by performing a post-hoc analysis using an assumption of a standard number of worms plated in each well (*n* = 20, where realistically this number ranged from 15 to 25). We estimated the % arrest based on this assumption, using a formula where the number of L1-L3 larvae = 20 –(true count of L4-adult) and found that 142 of the original 278 chemical wells were statistically different from random escapee wells (*Figure 2—figure supplements 2–3*). Given that 88% of those identified as statistically significant were in our prioritization group of 145, this provided us with confidence that they merited additional examination.

We re-tested the 145 chemicals in duplicate and found that 74 reproducibly suppressed the nemadipine-A-induced growth suppression of *unc-68* mutants (*Figure 2B*). Among these 74 'hits', we identified five wactives and seven non-wactives, 44 kinase inhibitors, and 18 compounds from the MicroSource library (*Supplementary file 1*). For several of the strongest suppressors, we performed dose-response analyses to confirm the effect of suppression (*Figure 2—figure supplement 4A*).

Many of the chemicals in our libraries have overlapping targets and/or functions (e.g. there are several EGFR inhibitors in the kinase library and several steroid hormones in the US Drug Collection). Therefore, we reasoned that hits that are overrepresented among groups of structurally and/or functionally related chemicals should be prioritized for further testing in zebrafish. To determine enrichment based on structural similarity, we compared the chemical fingerprints of the hits with those of the chemicals in each library by hierarchical clustering of their Tanimoto scores (*Figure 2C* and *Figure 2—figure supplement 5*; *Burns et al., 2015*). This method shows how structurally similar chemicals cluster with one another.

Using this methodology, we found that p38 inhibitors were significantly overrepresented (**p=0.0022, Fisher's exact test) when considering the total number of p38 inhibitors in the kinase inhibitor library. Interestingly, structurally dissimilar p38 inhibitors suppressed the phenotype (clusters 5, 6, 16, and 19 in *Figure 2C*), suggesting that inhibition of their common target was responsible for the activity. We then applied the same post-hoc analysis to the re-tested molecules as we did

with the primary screen to visualize changes in developmental stage distribution after chemical treatment (*Figure 2—figure supplement 4B–G*). As shown, p38 inhibitors were among the strongest suppressors identified from the inhibitor set. Finally, we validated p38 as a target by knocking down the p38 MAPK orthologs *pmk-1*, *pmk-2*, *pmk-3* alone and in combination, and we observed a modest increase in the proportion of *unc-68* mutants that escaped nemadipine-A induced growth arrest with each of the four conditions (*Figure 2D*).

Additionally, from the 44 hits in the kinase library we found that EGFR, PERK, and PLK1 inhibitors were overrepresented (**p=0.0060, *p=0.0109, *p=0.0326, respectively). Multiple GSK3 (n = 9) and PI3K (n = 3) inhibitors also strongly suppressed the phenotype, but these were not statistically overrepresented (p=0.4093 and p=0.1027, respectively). Among the 18 positive hits from the MicroSource library, several chemical classes were overrepresented (*Figure 2—figure supplement 5*): riboflavins (***p=0.0002), surfactants (***p=0.0002), benzophenones (**p=0.0018), anthracenes (*p=0.0279), salicylates (*p=0.0416), and tricyclic antihistamines (*p=0.0416). Interestingly, two DHPR inhibitors out of nine present in the library (**p=0.0063) were identified as suppressors of the growth arrest induced by nemadipine-A, itself a DHPR inhibitor. This may reflect competition for receptor binding which diminishes the effect of nemadipine-A, or perhaps a higher effective concentration which becomes agonistic to the receptor. Of note, riboflavin and riboflavin 5-phosphate sodium both strongly suppressed growth arrest (*Figure 2—figure supplement 4A,E*) and they appear to be structurally distinct from every other chemical in the US Drug Collection (cluster 11, *Figure 2—figure supplement 5*). Similarly, thiostrepton suppressed growth arrest strongly (*Figure 2—figure supplement 4E*), is a structurally unique molecule and overrepresented among the 18 hits (*p=0.0141). Altogether, the overrepresented groups of chemicals and the structurally unique molecules were prioritized for follow-up testing in *ryr1b* mutant zebrafish.

## Large-scale chemical screen in *ryr1* zebrafish

In parallel, we performed a screen in a zebrafish model of RYR1-RM. Zebrafish have two *ryr1* paralogs. Recessive mutations in *ryr1a* cause no overt phenotype, while recessive mutations in *ryr1b* result in abnormal swim behavior and early lethality after 11–13 days of life (*Hirata et al., 2007*). *ryr1a*; *ryr1b* double mutants exhibit no movement and have a median survival of 5 days of life (*Figure 3—figure supplement 1A–B*; *Chagovetz et al., 2019*). We used the double mutants for our screen because of their obvious motor phenotype and because variability in the *ryr1b* single mutant motor phenotype precluded large-scale screening (*Figure 3—figure supplement 1C*). We screened 436 kinase inhibitors at 10 µM from the DiscoveryProbe Kinase Inhibitor Library (ApexBio) and 1360 drugs at 10 µM from the US Drug Collection (MicroSource Discovery Systems), using improvement in motility of the *ryr1a*; *ryr1b* double mutants (as measured by touch-evoked escape response) as the primary outcome measure. We did not identify a single compound that was able to promote movement in these fish.

## Targeted drug testing in *ryr1* zebrafish

Using our prioritized list based on enrichment modeling, we next sought to determine if any positive hits from our *C. elegans* screen could improve phenotypes in either the *ryr1a*; *ryr1b* double mutants or the *ryr1b* single mutant zebrafish. We used motility as our outcome measure, examining both touch-evoked escape response and optogenetically induced swimming as per our previously established testing methodology (*Sabha et al., 2016*). Among the overall group of 74 hits, we tested 4/5 wactives, 7/7 non-wactives, 16/18 MicroSource drugs, and several inhibitors from the six classes of kinase inhibitors over-represented among the hits.

First, we tested all of these at a single concentration of 10 µM for the ability to promote movement in the *ryr1a*; *ryr1b* double mutants, which in the untreated state lack any movement. Consistent with our large-scale screen done with the *ryr1a*; *ryr1b* mutants, none of these chemicals improved the double mutant phenotype (*Figure 3*). We then tested all of these chemicals for the ability to suppress the abnormal touch-evoke escape response of *ryr1b* mutants, but these chemicals did not modulate this phenotype either. However, the utility of this phenotype as a readout of improvement may be unreliable given that it only transiently exists between 3–4 days of life, after which time the *ryr1b* mutants are indistinguishable by eye from their WT siblings (*Hirata et al., 2007*).

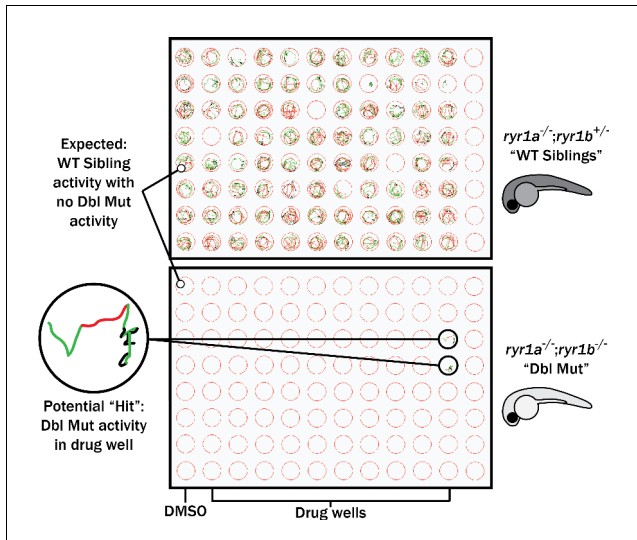

**Figure 3.** Schematic of our zebrafish screen methodology showing the expected motility of unaffected siblings ('WT Siblings') and double mutants ('Dbl Mut') and expected motility of immotile double mutants if a chemical suppressed the phenotype. We did not identify any suppressors of the double mutant phenotype.

The online version of this article includes the following figure supplement(s) for figure 3:

**Figure supplement 1.** Phenotypes of zebrafish *ryr1* mutants considered for chemical screening.

Hence, we assayed many of these chemicals on the *ryr1b* single mutants using our quantitative optogenetic motility assay. We tested for chemical-genetic interactions, which we defined as unexpected larval movement not resulting from the predicted combined effects of genotype and chemical treatment (see Methods). In other words, we assessed whether the movement speed of *ryr1b* mutants treated with a chemical was different from that which could be predicted from the combination of the *ryr1b* mutant effect plus the effect of the chemical on wild type. The p38 inhibitors SB239063 and PH-797804 decreased motility in wildtype controls, but had no effect on *ryr1* mutants (*Figure 4A–B*). The absence of effects in *ryr1* mutant models stands in contrast to predicted additive and multiplicative effects for these compounds (*Figure 4A'–B'*), suggesting that RyR1-related pathways mediate the effects of SB239063 and PH-797804. Additionally, we found significant positive interactions for three wactives, a PI3K inhibitor, and the anti-psoriatic drug anthralin (*Figure 4—figure supplement 1A–E*). We did not observe chemical-genetic interactions with two additional p38 inhibitors, SB203580 and SB202190 (*Figure 4C–D*) or with 18 other 'hits' (*Figure 4—figure supplements 1–3F–W*). Notably, we did not identify any chemicals that significantly improved *ryr1b* mutant movement speed relative to untreated mutant controls.

## Testing positive hits in C2C12 myotubes

We sought to examine the potential translatability of our findings to mammalian models of RYR1-RM. To accomplish this, we tested the effect of two p38 inhibitors on RyR1-dependent $Ca^{2+}$ release in C2C12 mouse myotubes. We examined this in wild type C2C12 cells and in a C2C12 *Ryr1* knockout line that we created using CRISPR/Cas9 gene editing. This new line contains a bi-allelic frameshift deletion mutation in *Ryr1* (which we refer to as 'KO'). Successful targeting of the *Ryr1* locus was demonstrated by Sanger sequencing, lack of off-target mutations verified by whole genome sequencing, and absence of RyR1 protein expression confirmed by western blot analysis (*Figure 5—figure supplement 1*).

We measured intracellular calcium release from RyR1 in response to acute application of 10 mM caffeine (*Tong et al., 1997*; *Meissner, 2017*) in wild type control and *Ryr1* KO C2C12 myotubes after 24 hr incubation with either SB203580 or SB202190. As expected, *Ryr1* KO myotubes treated with DMSO vehicle control lacked caffeine-induced $Ca^{2+}$ release (*Figure 5A–B*). KO myotubes treated with SB203580 or SB202190, however, exhibited a dose dependent increase in caffeine-

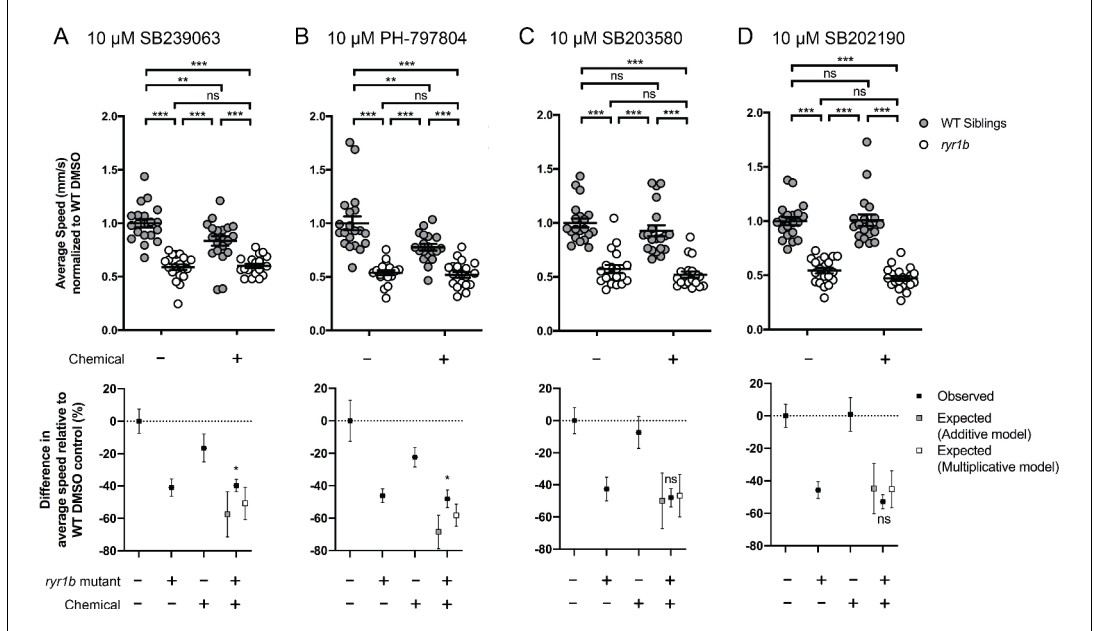

**Figure 4.** Positive chemical-genetic interactions between p38 inhibitors and *ryr1b* mutants were observed using zebrafish larval movement speed as a readout. Compared to the average speed of DMSO WT controls, treatment with (**A**) SB239063 and (**B**) PH-797804 reduced the average speed of WT siblings while the expected decrease in movement speed in treated *ryr1b* was not observed. The difference in average speed of *ryr1b*+p38 inhibitor compared to WT controls (i.e. WT siblings+DMSO vehicle) is higher than expected given the effects of genotype and chemical alone, indicative of a positive chemical-genetic interaction. Chemical-genetic interactions were not observed for p38 inhibitors (**C**) SB203580 and (**D**) SB202190. Data are presented as mean ± SEM movement speed (mm/s) normalized to DMSO-treated WT siblings for *n* = 2 independent experiments. Statistical analysis by two-way ANOVA followed by Tukey's multiple comparisons post-test. *p<0.05, **p<0.01, ***p<0.001; Sample size *n* = 19 or 20 for each treatment group in a set.

The online version of this article includes the following figure supplement(s) for figure 4:

**Figure supplement 1.** Chemical-genetic interactions were tested using zebrafish larval movement speed as a readout.

**Figure supplement 2.** Chemical-genetic interactions were tested using zebrafish larval movement speed as a readout.

**Figure supplement 3.** Chemical-genetic interactions were tested using zebrafish larval movement speed as a readout.

induced calcium release (*Figure 5A–B*). Conversely, these chemicals impaired caffeine-induced calcium release in control WT C2C12 myotubes (*Figure 5A–B*).

To interrogate the pathway specificity of SB203580 or SB202190, and to corroborate their positive effect, we examined caffeine-induced calcium release in the setting of siRNA knockdown of p38 isoforms α, β, and γ. Using commercially available siRNA we achieved roughly 50–80% knockdown of p38 MAPK targets as compared to non-targeting negative control siRNA (*Figure 5—figure supplement 2*). In *Ryr1* KO C2C12 cells, siRNA knockdown of *Mapk11* (p38β) promoted increased caffeine-induced calcium release in KO cells versus negative control siRNA (*Figure 5C*). Knockdown of *Mapk14* (p38α) and *Mapk12* (p38γ) also increased calcium release in *Ryr1* KO myotubes but to a lesser degree. These data thus suggest that in KO C2C12 cells, p38 inhibition (either via chemical or genetic inhibition) is able to promote intracellular calcium release independent of RyR1. This is consistent with the ability of p38 inhibitors and RNAi to suppress the *unc-68* (i.e. RyR1 null) phenotype in *C. elegans*.

Unlike with SB203580 or SB202190 treatment, siRNA knockdown of p38 isoforms did not impair $Ca^{2+}$ release in WT cells (*Figure 5C*), perhaps suggesting that inhibition may be partially caused by off-target effects of the chemicals. Alternatively, this may instead be a reflection of the incomplete p38 knockdown we achieved with siRNA, or could reflect siRNA toxicity, as caffeine-induced calcium release from WT cells was lower with control siRNA treatment when compared to DMSO-treated conditions.

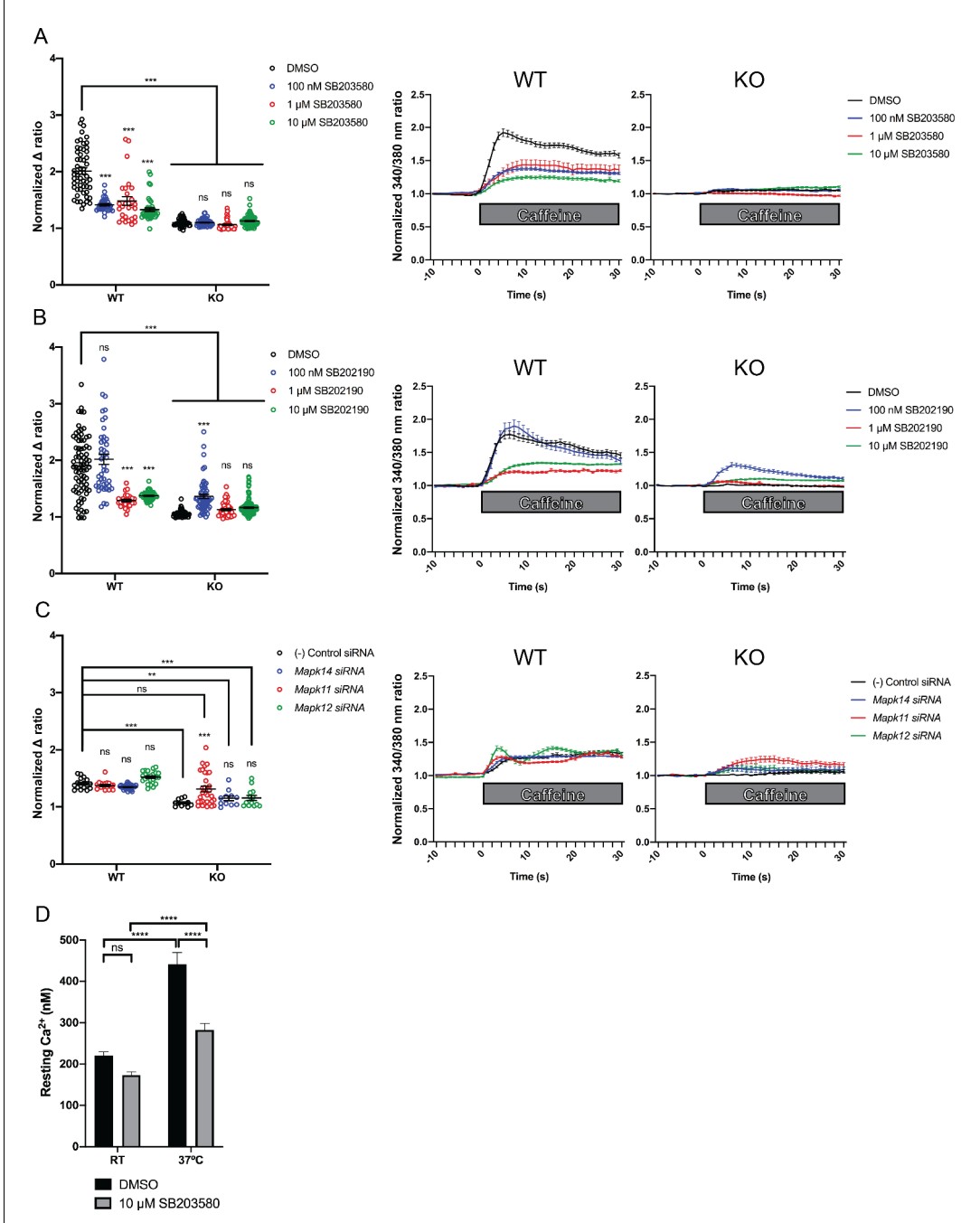

**Figure 5.** Intracellular calcium measurement in C2C12 myotubes. Ratiometric fura-2 imaging with 10 mM caffeine induction after treatment with p38 inhibitors (**A**) SB203580 (# of myotubes from left-right: $n$ = 56, 30, 28, 43, 45, 57, 46, 79) and (**B**) SB202190 ($n$ = 78, 44, 25, 45, 50, 58, 50, 142) or after (**C**) siRNA targeting p38α (*Mapk14*), p38β (*Mapk11*), and p38γ (*Mapk12*) ($n$ = 17, 27,18, 23, 11, 11, 33, 13). Panels on the left show the peak change in calcium concentration within 30 s of adding caffeine normalized to the average of 10 s of baseline resting free $Ca^{2+}$ concentration ($[Ca^{2+}]_i$), that is normalized Δ ratio, while plots on the right show calcium levels normalized to $[Ca^{2+}]_i$ after addition of caffeine. This data shows that p38 inhibition or knockdown impairs $Ca^{2+}$ release in wild type myotubes while promoting some $Ca^{2+}$ release in *Ryr1* knockout cells. Note: labels directly above each group indicate the statistical significance compared to DMSO control within the same genotype. Data are presented as mean ± SEM derived from $Ca^{2+}$ measurements of individual myotubes (*Figure 5—figure supplement 3A–C*). Corresponding plots showing $[Ca^{2+}]_i$ levels without normalization are also provided (*Figure 5—figure supplement 4D–F*). Statistical analysis by two-way ANOVA followed by Tukey's multiple comparisons post-test where *p<0.05, **p<0.01, ***p<0.001. (**D**) Overnight treatment of myotubes from Y522S mutant mice with 10 μM SB203580 significantly reduced a temperature-dependent increase in resting $Ca^{2+}$ concentration.

The online version of this article includes the following figure supplement(s) for figure 5:

*Figure 5 continued on next page*

**Figure supplement 1.** Characterization of *Ryr1* knockout (KO) C2C12 cells.

**Figure supplement 2.** Quantitative PCR analysis was performed to validate the efficiency of siRNA knockdown of three p38 MAPK genes in both WT and Ryr1 KO myotubes.

**Figure supplement 3.** Intracellular calcium measurement in C2C12 myotubes.

**Figure supplement 4.** Intracellular calcium measurement in C2C12 myotubes.

### SB203580 abrogates the temperature-dependent increase in resting Ca$^{2+}$ in YS myotubes

The fact that p38 chemical inhibitors negatively modulate swim behavior in WT zebrafish and caffeine-induced calcium release in C2C12 cells opens the possibility that they may serve as modifiers of phenotypes related to RyR1 hyperexcitability. To test this, we examined calcium dynamics in myotubes from the YS mouse model of RYR1-RM. The YS model contains a point mutation analogous to the Y522S mutation found in patients with malignant hyperthermia and central core pathology (*Quane et al., 1994*). The YS mutation enhances both the sensitivity of RyR1 to activators (e.g. DHPR, caffeine, 4-chloro-*m*-cresol) and the temperature-dependence of RyR1 Ca$^{2+}$ leak, alterations that underlie the MH susceptibility and exertional heat stroke phenotypes of these mice (*Chelu et al., 2006*; *Durham et al., 2008*; *Lanner et al., 2012*). We examined if the temperature dependent increase in resting myoplasmic Ca$^{2+}$ concentration in YS myotubes was abrogated by the p38 inhibitor SB203580. Overnight incubation of YS myotubes with 10 μM SB203580 significantly reduced the temperature-dependent increase in resting Ca$^{2+}$ observed in YS myotubes (*Figure 5D*).

## Discussion

In this study, we developed a multi-system pipeline for drug discovery and development for RYR1-RM. Using this platform, we were able to screen several thousand compounds and test their efficacy across multiple species. We identified p38 inhibitors as a new class of potential modifiers of RyR1 signaling. This platform can be applied to new compounds that may improve RYR1-RM phenotypes, as well as for additional diseases with suitable animal models.

The strengths of our pipeline include the rapidity with which we are able to screen drugs and the potential for increased translatability in drugs that positively modify a diverse group of in vivo models. In terms of speed, the initial screen in *C. elegans* was completed within two weeks, while the large-scale screen in zebrafish was accomplished in one month. While this does not approach the speed and scale of cell culture screens, it is fast and efficient and importantly is done in vivo using outcome measures that are relatable to human disease phenotypes. It would additionally be possible to add a cell culture based screen into the pipeline as a pre-screen or in parallel with the animal model testing. For example, studies for RyR1 functional interactors are underway using novel calcium indicators such as SERCaMP (*Henderson et al., 2014*); these indicators may prove ideal for screens in disease relevant cell models. Recently, a high-throughput screen was performed with HEK293 cells that successfully identified novel inhibitors of RyR1 (*Murayama et al., 2018*), supporting the potential utility of a cell-based screen in identifying modifiers of mutant RyR1 activity.

In terms of translatability, it is difficult to say whether a multi-organism strategy is superior to other approaches and/or more likely to yield targets that will work in humans. This is because, at present, no drugs have successfully been translated to patients with RYR1-RM. However, there is reason to speculate that a compound that can modify phenotype(s) associated with dysfunction of a gene in these diverse evolutionary settings may promote improvement in a more universal way.

Most relevant to future clinical intervention of RYR1-RM, our study identifies p38 inhibition as a modifier of RYR1-related phenotypes in both *C. elegans* and murine myotubes. The potential translational value of this observation awaits additional study, including in vivo testing in newly developed mouse models of RYR1-RM (see below). Interestingly, it was shown recently that combined treatment of *N*-acetylcysteine (NAC) and the p38 inhibitor SB203580 leads to robust expansion of myogenic satellite cell populations in vitro and in vivo (*L'honoré et al., 2018*). This is noteworthy because NAC has been shown to reduce aberrant oxidative stress and improve phenotypes of preclinical RYR1-RM models (zebrafish and patient cells) (*Dowling et al., 2012*). Based on these data,

NAC was recently tested via clinical trial in RYR1-RM patients, where a non-significant trend toward improvement was observed (*Todd et al., 2020*). These observations open the prospective of poly-therapy as a potential treatment strategy. Future work will be needed to establish the interplay between p38 inhibition and anti-oxidant therapy, both in the setting of normal myogenesis and with mutation in *RYR1*, and likely in the whole animal setting.

As p38 inhibition emerged as the most promising hit from our study, we considered potential mechanisms of action. Our observation of increased caffeine-induced intracellular calcium release in *Ryr1* KO C2C12 cells suggesting that p38 inhibition modulates non-RyR1 calcium channels. In the developing myotube, these could include RyR3 (*Tarroni et al., 1997*), the inositol 1,4,5-trisphosphate ($IP_3$) receptor ($IP_3R$), which has been shown to mediate calcium release in a caffeine-dependent manner (*Kang et al., 2010*), or the store-operated calcium entry (SOCE) system mediated by STIM1/ORAI1 (*Soboloff et al., 2006*; *Endo et al., 2015*). If we hypothesize a similar mechanism of action in both *C. elegans* and C2C12 cells, this would exclude RyR3 activation, as there is only a single RyR in *C. elegans* encoded by *unc-68*. Alternatively, both $IP_3R$ and the SOCE machinery are present in *C. elegans*, and enhanced calcium release from either of these sources provides a plausible explanation for our data. Interestingly, a previous study in endothelial cells showed that p38 inhibition (chemical and siRNA mediated) increased calcium entry via SOCE activation through a mechanism of blocking STIM1 phosphorylation (*Sundivakkam et al., 2013*). Future investigation, beyond the scope of this manuscript, is required to parse out the specific mechanisms related to the improvement seen with p38 inhibition.

An alternative explanation for some of our positive hits, is that they functioned by activating the dihydropyridine receptor (DHPR) and/or by overcoming its inhibition by nemadipine-A. This would be consistent with the observation that gain-of-function mutations in *egl-19*, the *C. elegans* homolog of DHPR, suppress the effects of nemadipine-A (*Kwok et al., 2006*) This mechanism could also explain the observed benefit of p38 inhibition in C2C12 myotubes, as DHPR has previously been shown to interact with and activated by RyR3 (*Sheridan et al., 2006*).

Of note, our data indicates that p38 chemical inhibitors may alter RyR1-mediated calcium dynamics in wild type zebrafish, wild type C2C12 cells, and YS mouse myotubes. Whether this is directly through p38 inhibition, or instead through other pathways, is not completely clear, as these chemicals may have off-target effects and our siRNA knockdown of p38 isoforms in wild type C2C12 cells did not replicate these effects. However, if we assume direct inhibition on the p38 pathway, plausible mechanisms include direct action on RyR1 (which has several phosphorylation sites known to impact its calcium release activity) or else modulation of RyR1 regulators (such as FKBP12, junctin or calsequestrin, proteins which bind to RyR1 and influence its channel function). To our knowledge, there is no evidence currently showing direct modulation of RyR1 phosphorylation status by the p38 pathway, nor any data supporting action on known RyR1 modifiers.

One shortcoming of our study is the relative lack of positive hits in the zebrafish. This may reflect the specific nature of our primary outcome measure in fish, which is focused on improving swimming locomotion. Locomotion is a complex trait that may be difficult to modify, especially in the double mutant fish which are completely paralyzed. We have observed mixed results with this outcome measure in other zebrafish myopathy models, where we have successfully identified chemicals that rescue the swimming defect in *mtm1* mutant zebrafish (*Sabha et al., 2016*), meanwhile we found no positive hits in *neb* deficient zebrafish which show complete paralysis (*Qiu et al., 2019*). As discussed below, we believe development of new models that do not completely lack RyR1 expression and that more closely mirror human mutations will overcome this limitation in the future.

One potential explanation for why we identified hits in *C. elegans* but not zebrafish is that the *C. elegans* screen was based on a developmental phenotype, and it is very plausible that alternative calcium pathways (e.g. DHPR, SOCE and/or IP3 receptor) are better able to compensate for developmental arrest as opposed to locomotor behavior, which is more specifically related to mature skeletal muscle function and likely has a more absolute requirement for RyR1-dependent ECC. This would be consistent with our positive results in our cell studies as well, which reflect immature/developing myotubes that may more readily utilize other calcium release pathways.

Another consideration of our study is that our screens were performed on models that completely lack expression of RyR1. There are currently no human patients with bi-allelic null mutations, and thus our models do not accurately mirror the human disease. In addition, our screens may miss chemicals that positively modulate mutant RyR1 channels but that cannot compensate for its

complete loss. One of our primary future directions is therefore to develop new models of RYR1-RM that are better suited to drug discovery and for subsequent testing in mammals. In this vein, there are very recent publications detailing new recessive RYR1-RM mouse models including a p.G2435R mutant (*Lopez et al., 2018*), a p.A4329D/p.Q1970fsX16 compound heterozygote model (*Elbaz et al., 2019*), and a p.T4706M/indel compound heterozygote model generated by our group (*Brennan et al., 2019*). In addition, seven RYR1-RM equivalent mutations in *unc-68* were modeled by transgenic overexpression in *C. elegans* and these mutants exhibited hypersensitivity to caffeine and the malignant hyperthermia triggering agent halothane (*Baines et al., 2017*).

## Conclusion

We established a unique 'multi species' pipeline for drug discovery for RYR1-RM. This platform is rapid and robust, and provides the ability to examine multiple different types of in vivo models. Our study lays the ground work for its future use in RYR1-RM drug development, and for establishment of similarly platforms for other rare diseases.

# Materials and methods

**Key resources table**

| Reagent type (species) or resource | Designation | Source or reference | Identifiers | Additional information |
|---|---|---|---|---|
| Genetic reagent (*C. elegans*) | *unc-68*(r1162) | *C. elegans* Genetics Center (University of Minnesota) | Strain TR2171 RRID:WB-STRAIN:WB Strain00034950 | |
| Genetic reagent (*C. elegans*) | *unc-68*(r1161) | *C. elegans* Genetics Center (University of Minnesota) | Strain TR2170 RRID:WB-STRAIN:WB Strain00034949 | |
| Strain, strain background (*E. coli*) | HB101 | *C. elegans* Genetics Center (University of Minnesota) | | |
| Strain, strain background (*E. coli*) | HT115 (DE3) | *C. elegans* RNAi Collection (Ahringer) from Source BioScience | Clones IV-9P08 (*pmk-1*), IV-4G23 (*pmk-2*), IV-4I01 (*pmk-3*), and I-1K04 (*pop-1*) | RNAi library |
| Genetic reagent (*D. rerio*) | *ryr1a*(z42) | *Chagovetz et al., 2019* | ZFIN ID: ZDB-ALT-180925–10 | |
| Genetic reagent (*D. rerio*) | *ryr1b*(mi340) | *Hirata et al. (2007)* | ZFIN ID: ZDB-ALT-070928–1 | |
| Chemical compound, drug | The US Drug Collection | MicroSource Discovery Systems, Inc | | Chemical library |
| Chemical compound, drug | DiscoveryProbe Kinase Inhibitor Library | APExBIO | Catalog #L1024 | Chemical library |
| Chemical compound, drug | Nemadipine-A | ChemBridge | Catalog #5619779 | |
| Chemical compound, drug | Optovin analog 6b8 | ChemBridge | Catalog #5707191 | |
| Chemical compound, drug | SB203580 | Sigma | Catalog #S8307 | |
| Chemical compound, drug | SB202190 | Sigma | Catalog #S7067 | |
| Cell line (*M. musculus*) | C2C12 myoblasts | ATCC | Catalog #CRL1772 | |
| Cell line (*M. musculus*) | *Ryr1* KO C2C12 myoblasts | This paper | | 2 bp deletion in exon 6 of *Ryr1*: c.497_498del, p.Val166GlyfsX3 |

*Continued on next page*

*Continued*

| Reagent type (species) or resource | Designation | Source or reference | Identifiers | Additional information |
|---|---|---|---|---|
| Antibody | anti-Ryanodine receptor (Mouse monoclonal) | Developmental Studies Hybridoma Bank (University of Iowa) | Catalog #34C, RRID:AB_528457 | WB (1:100) |
| Antibody | anti- beta actin (Mouse monoclonal) | Abcam | Catalog #ab8226 RRID:AB_306371 | WB (1:5000) |
| Sequence-based reagent | ON-TARGETplus Mouse Mapk11 siRNA | Horizon Discovery (Dharmacon) | Catalog # J-050928-05-0010 | Target sequence: 5'-AUGAGGAGAUGACCGGAUA-3' |
| Sequence-based reagent | ON-TARGETplus Mouse Mapk12 siRNA | Horizon Discovery (Dharmacon) | Catalog # J-062913-05-0010 | Target sequence: 5'-AAUGGAAGCGUGUGACUUA-3' |
| Sequence-based reagent | ON-TARGETplus Mouse Mapk14 siRNA | Horizon Discovery (Dharmacon) | Catalog # J-040125-06-0010 | Target sequence: 5'-GCAAGAAACUACAUUCAGU-3' |
| Sequence-based reagent | ON-TARGETplus Non-targeting siRNA #1 | Horizon Discovery (Dharmacon) | Catalog # D-001810-01-05 | Target sequence: 5'-UGGUUUACAUGUCGACUAA-3' |
| Sequence-based reagent | Tbp forward primer | This paper | Mouse *Tbp* gene: ENSMUSG00000014767 | 5'-TGCTGCAGTCATCATGAG-3' |
| Sequence-based reagent | Tbp reverse primer | This paper | Mouse *Tbp* gene: ENSMUSG00000014767 | 5'-CTTGCTGCTAGTCTGGATTG-3' |
| Sequence-based reagent | Mapk11 forward primer | This paper | Mouse *Mapk11* gene: ENSMUSG00000053137 | 5'-CCAGCAATGTAGCGGTGAACGAG-3' |
| Sequence-based reagent | Mapk11 reverse primer | This paper | Mouse *Mapk11* gene: ENSMUSG00000053137 | 5'-GCATGATCTCTGGCGCCCGGTAC-3' |
| Sequence-based reagent | Mapk12 forward primer | This paper | Mouse *Mapk12* gene: ENSMUSG00000022610 | 5'-CACTGAGGATGAACCCAAGGCC-3' |
| Sequence-based reagent | Mapk12 reverse primer | This paper | Mouse *Mapk12* gene: ENSMUSG00000022610 | 5'-CTCCTAGCTGCCTAGGAGGCTTG-3' |
| Sequence-based reagent | Mapk14 forward primer | This paper | Mouse *Mapk14* gene: ENSMUSG00000053436 | 5'-CAGCAGATAATGCGTCTGACGGG-3' |
| Sequence-based reagent | Mapk14 reverse primer | This paper | Mouse *Mapk14* gene: ENSMUSG00000053436 | 5'-GCGAAGTTCATCTTCGGCATCTGG-3' |
| Commercial assay or kit | RNeasy Mini Kit | Qiagen | Catalog #74104 | |

## Animal ethics statement

All zebrafish experiments were performed in accordance with all relevant ethical regulations, specifically following the policies and guidelines of the Canadian Council on Animal Care and an institutionally reviewed and approved animal use protocol (#41617). No additional ethical approval was required for our experiments with the invertebrate nematode worm *C. elegans*.

## Chemical sources

Chemical libraries used in this study include the US Drug Collection (1280 compounds; MicroSource Discovery Systems Inc), the DiscoveryProbe Kinase Inhibitor Library (436 compounds; APExBIO), the GlaxoSmithKline Published Kinase Inhibitor (PKI) Set (880 compounds; William Zuercher), and a collection of 770 worm-bioactives ('wactives') that were identified in a screen for bioactive small molecules in *C. elegans* (*Burns et al., 2015*). Nemadipine-A (#5619779), optovin analog 6b8 (#5707191), and wactives/non-wactives (ID numbers in *Supplementary file 1*) were purchased from ChemBridge. Chemicals from the US Drug Collection identified as 'hits' in the *C. elegans* screen were purchased from Sigma-Aldrich for testing in zebrafish. Kinase inhibitors representative of those identified as 'hits' in the *C. elegans* screen, including p38 inhibitors, were purchased individually from APExBIO or selected from the DiscoveryProbe Kinase Inhibitor Library for testing in zebrafish and cell lines.

## *C. elegans* strains and culture

All animals were cultured under standard methods at 20°C (*Burns et al., 2015*). The wild type (N2), and *unc-68(r1161)* and *unc-68(r1162)* (TR2170 and TR2171, respectively) strains of *Caenorhabditis*

*elegans* were obtained from the *C. elegans* Genetics Center (University of Minnesota). The 'thrashing assay' is performed by counting the waveforms propagated by individual *C. elegans* in one minute as previously described (*Maryon et al., 1996*).

## *C. elegans* chemical screening

The protocol for the 96-well liquid-based chemical screens was described previously (*Burns et al., 2015*). Briefly, nematode growth media (NGM; for recipe see *Burns et al., 2015*) was used to concentrate saturated *E. coli* HB101 bacteria two-fold (NGM-HB101). Nemadipine-A (NEM) was added to NGM+HB101 to a final concentration of 31.25 µM/0.5% DMSO (NEM+NGM+HB101). A total of 40 µL of NEM+NGM+HB101 was dispensed into each well of a 96-well plate, and 300 nL of chemical dissolved in DMSO was pinned into the wells using a 96-well pinning tool (V and P Scientific). At Day 0, approximately 20 synchronized first larval-stage (L1), *unc-68*(*r1162*) (TR2171) worms obtained from an embryo preparation were added to each well in 10 µL of M9 buffer (*Burns et al., 2015*). The *unc-68(r1162)* strain TR2171 was selected for screening because its growth rate was comparable to N2 wild type whereas strain TR2170 grew noticeably slower than N2. The final concentration of dimethyl sulfoxide (DMSO) in the wells was 1% v/v. Plates were sealed with parafilm, wrapped in damp paper towels to reduce evaporation in wells, and incubated for 6 days at 20℃ while shaking at 200 rpm (New Brunswick I26/I26R shaker, Eppendorf). A stereomicroscope was used to assess developmental stage after 6 days incubation. All preliminary screens and re-tests were performed in duplicate. Post-hoc statistical analysis was performed by assuming 20 worms in each well and assigning each individual worm in L1-L3, L4, or adult stage a rank score of 1, 2, and 3, respectively. Next, the developmental stage distributions were compared to the random escapee wells using a nonparametric Kruskal-Wallis test with Dunn's multiple comparisons post-test in GraphPad Prism 8.

## *C. elegans* RNA interference

RNAi knock-down was carried out in 96-well plate liquid culture as previously described (*Lehner et al., 2006*). Synchronized L1-stage *unc-68(r1162)* mutant worms were fed *E. coli* HT115 bacteria expressing double-stranded RNA targeting *pmk-1, pmk-2, pmk-3* or a combination of the three from the Source BioScience RNAi library. The *E. coli* HT115 strain carrying the L4440 empty vector was used as a control for RNAi machinery induction, and *pop–1* RNAi which produces a severe embryonic lethality phenotype was used as a control for RNAi induction efficiency. A total of 40 µL of bacterial suspension containing nemadipine-A or DMSO was dispensed into each well of a flat-bottomed 96-well plate. Approximately 25 *unc-68(r1162)* L1 worms were dispensed into the wells in 10 µL of M9 buffer. The final concentration of nemadipine-A in the wells was 25 µM in 0.4% DMSO v/v. Plates were sealed with parafilm and incubated at 20℃ with shaking at 200 rpm. On Day six the plates were observed under a dissection microscope and the distribution of developmental stages in each condition was assessed. Each experiment was performed in quadruplicate and repeated at least three times. For statistical analysis, each individual worm in L1-L3, L4, or adult stage was assigned a rank score of 1, 2, and 3, respectively, and the developmental stage distributions were compared to the negative empty vector control using a nonparametric Kruskal-Wallis test with Dunn's multiple comparisons post-test in GraphPad Prism 8.

## Cheminformatics

Pairwise Tanimoto coefficient scores were calculated for each chemical 'hit' and compound in the screening library using OpenBabel (http://openbabel.org) as previously described (*Burns et al., 2015*). Tanimoto coefficient scores were hierarchically clustered in Cluster 3.0 using an unweighted Euclidean distance similarity metric with complete linkage clustering and visualized in TreeView as previously described (*Folts et al., 2016*). For enrichment analysis, chemicals were counted based on the number of structurally similar members in each cluster (Tanimoto scores were >0.55 for the majority of members in a given cluster) and Fisher's exact test (GraphPad Prism 8) was used to calculate enrichment.

## Zebrafish care and husbandry

In this study, we used *ryr1a* and *ryr1b* mutant alleles that have been previously characterized (*Hirata et al., 2007*; *Chagovetz et al., 2019*). Both mutants result in loss of RYR1 protein expression

from the mutant allele. For follow-up screens, we generated single $ryr1b^{-/-}$ mutants via incross of $ryr1b^{+/-}$ carriers. Phenotypic analysis of all $ryr1$ mutants was performed on a stereomicroscope.

## Zebrafish chemical treatments

All chemical stocks were prepared in DMSO and added to egg water at 0.1–0.5% of the final volume to prepare working concentrations (depending on chemical solubility). Equal volumes of vehicle solvent were used in all conditions for a single assay. Note that methylene blue was not added to the egg water. Dishes or 96-well plates were sealed with parafilm, wrapped in aluminum foil, and incubated at 28.5°C until the assay date. Different volumes and culture dish formats were used depending on the endpoint assay.

## Zebrafish chemical screening

The US Drug Collection (1280 compounds) and the DiscoveryProbe Kinase Inhibitor Library (436 compounds) were screened for chemicals that could promote motility in $ryr1a; ryr1b$ double mutants. Library stocks of 10 mM in DMSO were added to 0.1% of the final volume to egg water to prepare a 10 µM screening concentration. Specifically, 150 µL of egg water was added to every well of two separate 96-well plates. Double mutants were generated by in-cross of $ryr1a^{-/-};ryr1b^{+/-}$ mutants. Embryos were manually dechorionated at one dpf. Then at two dpf two double mutant embryos were added to each well of one plate while two phenotypically wild type siblings were added to the second plate as a control. We reasoned that testing two larvae would be sufficient to detect if a chemical rescued the striking immotile phenotype. Care was taken to lower embryos to the bottom of a Pasteur pipette tip and deposit embryos into the well by surface tension to minimize changes to well volume. Next, 250 µL of the drug library was prepared at 40 µM working concentration in a separate 96-well plate by adding 1 µL of the 10 mM stock to 249 µL of egg water. Next, 50 µL of the 40 µM working concentration was added to the 150 µL water containing embryos to give a final concentration of 10 µM drug. After 24 hr incubation in chemical, motility of 3 dpf wild type and double mutant larvae was assessed by touch-evoked response (*Hirata et al., 2007*) under a stereomicroscope.

## Photoactivation of motor behavior assay

At three dpf, $ryr1b$ mutants were segregated from wild type ($ryr1b^{+/+}$ or $ryr1b^{+/-}$) siblings based on their phenotype and distributed into sterile 6 cm tissue culture dishes containing 10 mL of egg water plus chemical. Please note that care was taken to evenly distribute larvae from one pair of parents across all conditions, that is sibling-matching, to minimize potential effects arising from clutch-to-clutch variability. Additionally, dysmorphic or underdeveloped larvae were not used. All assays were performed at 4 dpf after 24 hr incubation using the ZebraBox platform (ViewPoint) and 10 µM opto-vin analog 6b8 as previously described (*Sabha et al., 2016*). Using G*Power Version 3.1 (*Faul et al., 2009*), we calculated that a sample size $n = 9$ larvae per group would allow us a 99% probability of detecting a difference between WT and $ryr1b$ mutants given their group means and standard deviations in movement speed (mm/s). For the majority of chemicals, two independent experiments with $n = 9$ or 10 larvae per group were performed. To compare effects from independent experiments, movement speed of individual larvae in each group were normalized to the average movement speed of DMSO treated WT siblings from the same assay. This would allow us to estimate chemical-genetic interactions based on an unexpected change in the difference between WT and $ryr1b$ following chemical treatment. We adapted the formulas for calculating additive and multiplicative models of genetic interactions Figure 1 in *Baryshnikova et al. (2013)* to calculate and visualize expected chemical-genetic interactions based on larval movement speed. Statistical significance for the interaction was calculated by two-way ANOVA with Tukey's multiple comparison post-test (GraphPad Prism 8).

## Generating *Ryr1* knockout C2C12 cells using CRISPR-Cas9 strategy

The original C2C12 (ATCC CRL1772) was purchased from American Type Culture Collection (Manassas, VA, USA). The sgRNA sequence (5'-AGGAGAGAAGGTTCGAGTTG-3') against *Ryr1* in Exon six was designed by the online CRISPR Design Tool (http://tools.genome-engineering.org) and cloned at the BbsI site into pSpCas9(BB)−2A-Puro (PX459) V2.0 (Addgene plasmid ID: 62988). The CRISPR

plasmids are transfected into C2C12 cells by electroporation (Amaxa Nucleofector Lonza). Seventy-two hours later, the transfected cells were selected in medium containing 4 µg/mL puromycin for three days and then subcloned into 96-well plates. Once at sufficient cell density, the genomic DNA of subclones was analyzed by Sanger sequencing. Primers used were (forward) 5'-GTGTGACGG-GAGTCCCAAAT-3' and (reverse)5'-ACTGGGCATGCCAATGATGA-3'. Cells were tested and found negative for mycoplasma.

## Western blot

Protein was isolated in RIPA buffer from C2C12 myotubes at 5 days after starting differentiation. Cells are incubated with 100 nM SB202190, 10 µM SB203580, DMSO and without treatment for 24 hr from day4. A total of 30 µg of total protein was run on either 4.5% SDS-acrylyamide gel for RyR1 or 15% SDS-acrylamide gel for β-actin. Blots were run for 3–3.5 hr for Ryr1 or for 2–2.5 hr for βactin at 100 V and transferred overnight at 20 V. The membrane was blocked in 3% bovine serum albumin (BSA) in Tris Buffered Saline with Tween 20 (TBST) for 1 hr at room temperature before incubating with primary antibodies overnight at 4°C. Antibodies used were anti-Ryanodine receptor antibody 34C (Developmental Studies Hybridoma Bank) at 1:100 dilution and anti-beta actin antibody (Abcam) at 1:5000 dilution. After three washes in TBST, blots were incubated with Anti-Mouse IgG-HRP conjugate (Bio-Rad) at 1:10000 dilution. Blots were imaged by chemiluminescence (Western Lightning Plus-ECL, PerkinElmer) using the Gel Doc XR + Gel Documentation System (BioRad), and band signal intensities determined using ImageLab software (BioRad).

## Small-interfering RNA (siRNA) interference in myotubes

C2C12 myoblasts were seeded at $2.5 \times 10^4$ cells·cm$^{-2}$ density in 24-well plates (Falcon) containing glass coverslips coated with 5 µg/cm$^2$ collagen and grown in DMEM with 20% (v/v) fetal bovine serum (FBS). Once cells reached 80–90% confluency, media was changed to differentiation media (DMEM with 2% (v/v) horse serum, 1 µg/mL insulin and 50 µg/mL gentamicin) representing Day 0 of differentiation. At Day 4, myotubes were transfected for 6 hr with 50 pmol ON-TARGETplus siRNA (Horizon Discovery) against p38α/*Mapk14* (5'-GCAAGAAACTACATTCAGT-3'), p38β/*Mapk11* (5'-A TGAGGAGATGACCGGATA-3'), p38γ/*Mapk12* (5'-AATGGAAGCGTGTGACTTA-3') or a non-target-ing negative control (5'-TGGTTTACATGTCGACTAA-3') in complex with 5 µL Lipofectamine RNAi/MAX (Invitrogen) in 500 µL Optimem solution (Gibco). Fresh differentiation medium was added until Day five when myotubes were transfected again following the same protocol. Finally, myotubes were maintained in differentiation media until Day 6. At Day 6, myotubes were either used for calcium measurements or qPCR analysis. It is important to note that siRNA knockdown was started after differentiation into myotubes because siRNA knockdown of either the α, β, or γ isoforms of p38 prevents C2C12 myoblast differentiation into myotubes (*Wang et al., 2008*).

## Quantitative real-time PCR

RNA was extracted with the RNeasy Mini kit (Qiagen) from $n = 3$ independent myotube cultures for each siRNA condition. Each sample was run in triplicate using SYBR Green master mix Applied Bio-systems StepOne system (ThermoFisher). Comparative $\Delta\Delta C_T$ method was used to determine relative expression of target genes after siRNA knockdown. Briefly, target gene expression was normalized to endogenous control gene *Tbp* as an endogenous control to obtain $\Delta C_T$ values. Next, relative fold change in gene expression for each siRNA condition was calculated by normalizing to the mean $\Delta C_T$ value for negative control siRNA. The following primer pairs were used: *Tbp* (Forward: 5'-TGC TGCAGTCATCATGAG-3'; Reverse: 5'-CTTGCTGCTAGTCTGGATTG-3'), p38α/*Mapk14* (Forward: 5'-CAGCAGATAATGCGTCTGACGGG-3'; Reverse: 5'-GCGAAGTTCATCTTCGGCATCTGG-3'), p38β/*Mapk11* (Forward: 5'-CCAGCAATGTAGCGGTGAACGAG-3'; Reverse: 5'-GCATGATCTC TGGCGCCCGGTAC-3'), p38γ/*Mapk12* (Forward: 5'-CACTGAGGATGAACCCAAGGCC-3'; Reverse: 5'-CTCCTAGCTGCCTAGGAGGCTTG-3').

## Intracellular Ca$^{2+}$ measurements in myotubes

Intracellular Ca$^{2+}$ measurements were obtained from Fura-2 (Invitrogen) AM-loaded myotubes as described previously (*Goonasekera et al., 2007*). Briefly, myotubes were differentiated for 5–6 days on glass bottom dishes and loaded with 5 µM Fura-2 AM for 45 min at 37°C in a normal rodent

Ringer's solution consisting of 145 mM NaCl, 5 mM KCl, 2 mM $CaCl_2$, 1 mM $MgCl_2$, 10 mM HEPES, pH 7.4. Coverslips of Fura-2–loaded cells were then mounted in a tissue chamber on the stage of an epifluorescence-equipped inverted microscope (Zeiss). Cells were sequentially excited at 340- and 380 nm wavelength and fluorescence emission at 510 nm was collected using a high-speed CCD camera (Hamamatsu). The results are presented as the ratio of 340/380 nm. Maximal increase or peak change in intracellular $Ca^{2+}$ by induction of 10 mM caffeine was defined as the difference between peak and 10 s of baseline fluorescence ratios prior to addition of caffeine. To better visualize differences in peak change across multiple treatment groups, 340/380 ratios for each individual myofiber were normalized to their own 10 s of baseline 340/380 ratios prior to addition of caffeine. Data are also presented without normalization in *Figure 5—figure supplement 1D–F*.

### Temperature-dependence of resting calcium in YS myotubes

Resting $[Ca^{2+}]_i$ was measured in primary myotubes derived from *Ryr1$^{Y524S/+}$* (YS) mice cultured on glass bottom dishes using Fura-2 (Invitrogen) and a temperature controlled chamber (*Lanner et al., 2012*). The day before measurements, myotube cultures were incubated overnight in either vehicle or 10 µM SB203580. Fura-2 was excited at 340 nm and 380 nm and images of myotubes were collected using a CCD camera connected to a TILL monochromator (TILL Photonics Inc). The myotubes were first imaged at temperature just above room temperature (25˚C). The temperature of the bathing solution was then raised to 37˚C and the same cells were re-imaged again at 37˚C for paired statistical comparison. Ratio images of 340/380 nm for Fura-2 were created using TILLvisION software and analyzed offline using ImageJ software. Resting free calcium concentration ($[Ca^{2+}]_i$) was calculated using a calibration curve of Fura-2 as described previously (*Lanner et al., 2012*).

### Statistical analysis

Tests of statistical significance were performed using Microsoft Office Excel 2008 (Microsoft) and GraphPad Prism eight for Mac OSX (GraphPad Software). Differences were considered to be statistically significant at $p < 0.05$ (*), $p < 0.01$ (**), or $p < 0.001$ (***). All data unless otherwise specified are presented as mean $\pm$ SEM.

## Acknowledgements

The authors wish to thank Dr. Andrew Burns at University of Toronto for helpful discussion and insight. We wish to thank Dr. David Grunwald at University of Massachusetts Medical School for generously providing the *ryr1a* zebrafish mutants. The study was funded by grants from Muscular Dystrophy Association and the RYR1 Foundation (JJD and RTD).

## Additional information

### Funding

| Funder | Grant reference number | Author |
|---|---|---|
| Muscular Dystrophy Association | | Robert T Dirksen<br>James J Dowling |
| RYR-1 Foundation | | Robert T Dirksen<br>James J Dowling |
| Canadian Institutes of Health Research | 363863 | Robert T Dirksen<br>James J Dowling |

The funders had no role in study design, data collection and interpretation, or the decision to submit the work for publication.

### Author contributions

Jonathan R Volpatti, Conceptualization, Formal analysis, Investigation, Methodology, Writing - original draft, Writing - review and editing; Yukari Endo, Robert T Dirksen, Formal analysis, Investigation, Methodology, Writing - review and editing; Jessica Knox, Investigation, Methodology, Writing - review and editing; Linda Groom, Stephanie Brennan, Ramil Noche, Investigation; William J

Zuercher, Resources; Peter Roy, Conceptualization, Resources, Supervision, Project administration, Writing - review and editing; James J Dowling, Conceptualization, Resources, Formal analysis, Supervision, Funding acquisition, Methodology, Writing - original draft, Project administration, Writing - review and editing

## Author ORCIDs

Robert T Dirksen 🔟 http://orcid.org/0000-0002-3182-1755
James J Dowling 🔟 https://orcid.org/0000-0002-3984-4169

## Ethics

Animal experimentation: All zebrafish experiments were performed in accordance with all relevant ethical regulations, specifically following the policies and guidelines of the Canadian Council on Animal Care and an institutionally reviewed and approved animal use protocol (#41617). No additional ethical approval was required for our experiments with the invertebrate nematode worm *C. elegans*.

## Decision letter and Author response

Decision letter https://doi.org/10.7554/eLife.52946.sa1
Author response https://doi.org/10.7554/eLife.52946.sa2

## Additional files

### Supplementary files

• Source data 1. Data files for the *C. elegans* chemical screen, zebrafish motility assays, and myotube calcium measurements.

• Supplementary file 1. List of 74 chemicals that suppressed the synthetic, nemadipine-A-induced *unc-68* growth arrest.

• Supplementary file 2. List of individual chemicals used in this study. This list does not include all of the chemicals present in the chemical libraries screened.

• Transparent reporting form

### Data availability

All data generated or analysed during this study are included in the manuscript and supporting files. Source files are available for all figures.

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
