## [Decision Letter]

**Acceptance summary:**

Modern "target-based" drug discovery is a costly, time-consuming, and frankly inefficient endeavor, with new drugs now costing billions of dollars. This paper exemplifies how screening drugs directly in living miniaturized disease models (e.g., worms, fish, and mammalian cell culture) has the potential to streamline the discovery process and reveal promising new drug targets. Using a novel cross-species screening cascade, the authors implicate the p38 signaling pathway in mediating ryanodine receptor type I-related myopathies (RYR1-RMs), suggesting that p38 inhibition could provide a new therapeutic strategy for treating this devastating childhood muscle disease for which no current treatments are available.

**Decision letter after peer review:**

Thank you for submitting your article "Identification of drug modifiers for RYR1 related myopathy using a multi-species discovery pipeline" for consideration by *eLife*. Your article has been reviewed by four peer reviewers, including Jeff S Mumm as the Reviewing Editor and Reviewer #1, and the evaluation has been overseen by Didier Stainier as the Senior Editor. The following individuals involved in review of your submission have agreed to reveal their identity: Isaac Pessah (Reviewer #2); Guy M Benian (Reviewer #3); Clarissa Henry (Reviewer #4).

The reviewers have discussed the reviews with one another and the Reviewing Editor has drafted this decision to help you prepare a revised submission.

Summary:

The manuscript reports results from extensive whole organism chemical screens in worms and fish, as well as a mouse cell culture models, of a debilitating muscle disease associated with mutations in the ryanodine receptor type I-related (RYR1) gene, specifically: (a) Mitigation of developmental arrest in RyR mutant nematodes (aka, "unc68") exposed to nemadipine-A; (b) Mitigation of motor function deficits in zebrafish models with and without expression of ryr1 isoforms (WT and Ca-leaky RYR1 mutations), and; (c) An in vitro myotube model that assesses mitigation of ER/SR Ca leak due to expression of ryanodine receptor (RyR) pathogenic mutations. The pipeline the authors developed is a substantial achievement and exemplifies the potential for cross-species phenotypic screening to enhance modern drug discovery. The authors finding that p38 inhibition could provide a new therapeutic strategy for treating ryanodine receptor type I-related myopathies (RYR1-RMs) is intriguing and deserving of further study. Overall, the study is innovative and enlightening. However, several key issues were flagged by the reviewer's which need to be addressed to improve the quality of the manuscript.

Essential revisions:

1) Power calculations should be performed and confirmatory tests of positive hits performed with adequate samples sizes to overcome shortcomings of the primary screen regarding inadequate sample size.

2) Given the high concentrations of compounds used in the *C. elegans* assays (sometimes >4 orders of magnitude higher than published affinities for target), target dependency of p38 inhibitor effects in *C. elegans* should be validated using RNAi. Similarly, siRNA should be used to validate p38 as a target in C2C12 myotube cultures. Alternatively, dose-responses could be performed to better define concentrations range producing observed effects.

3) "…loss of expression mutation…" for the *unc-68* allele used, *unc-68*(r1162), is not supported. A Western Blot to showing loss of protein is needed to confirm. If not a null allele this could explain differences between worm and fish screens (and be useful for addressing any discrepancies with requested RNA0 and siRNA experiments in #1).

---

## [Author Response]

Essential revisions:1) Power calculations should be performed and confirmatory tests of positive hits performed with adequate samples sizes to overcome shortcomings of the primary screen regarding inadequate sample size.

Prior to our zebrafish drug testing, we performed a theoretical power calculation, the results of which supported the sample sizes used in our large-scale screen and in our experiments to test positive hits (Materials and methods, subsection “Photoactivation of motor behavior assay”). Additionally, we performed post-hoc statistical analyses on the primary *C. elegans* screen (subsection “Large-scale chemical screen in *C. elegans* identifies 74 *unc-68* suppressors”, fourth paragraph) and on our re-testing (seventh paragraph of the aforementioned subsection), as well as follow-up dose-response testing in worms for a select number of strong suppressors from the screen (Figure 2—figure supplement 4). The post-screen testing and analysis provides assurance that effects from chemicals from the primary screen were true positive hits. This is outlined in detail in the manuscript.

2) Given the high concentrations of compounds used in the *C. elegans* assays (sometimes >4 orders of magnitude higher than published affinities for target), target dependency of p38 inhibitor effects in *C. elegans* should be validated using RNAi. Similarly, siRNA should be used to validate p38 as a target in C2C12 myotube cultures. Alternatively, dose-responses could be performed to better define concentrations range producing observed effects.

RNAi against the three p38 orthologs (*pmk-1, pmk-2*, and *pmk-3*) was tested in wildtype N2 and *unc-68*(r1162) *C. elegans* in the presence and absence of nemadipine-A, using developmental arrest as a readout (see Materials and methods and subsection “Large-scale chemical screen in *C. elegans* identifies 74 *unc-68* suppressors”, seventh paragraph). We observed that p38 knockdown significantly increased the proportion of L4/adult stage *unc-68* worms versus empty vector control (Figure 2). Similarly, siRNA knockdown of *Mapk14* (p38α), *Mapk11* (p38β), and *Mapk12* (p38γ) was performed in wildtype and *Ryr1* knockout C2C12 myotubes (see Materials and methods and subsection “Testing positive hits in C2C12 myotubes”, third and last paragraphs). The knockdown efficiency was validated by qPCR (Figure 5—figure supplement 2) and effect of knockdown on caffeine-induced intracellular calcium release was assessed (Figure 5C). We observed similar effects to chemical inhibitor treatment after knockdown of *Mapk11* (p38β), where calcium concentrations were slightly elevated in Ryr1 KO vs. control siRNA (Figure 5C and the aforementioned paragraphs).

3) "…loss of expression mutation…" for the unc-68 allele used, unc-68(r1162), is not supported. A Western Blot to showing loss of protein is needed to confirm. If not a null allele this could explain differences between worm and fish screens (and be useful for addressing any discrepancies with requested RNA0 and siRNA experiments in #1).

A Western Blot confirming the loss of protein expression in *unc-68*(r1162) is found in Figure 5 of Maryon et al., 1998. The r1162 allele is a large deletion that spans the RyR1 calcium channel pore. Based on this existing published Western blot, the mutation results in complete loss of RyR1 protein expression and thus is considered a functional null. We updated the first paragraph of the subsection “Large-scale chemical screen in *C. elegans* identifies 74 *unc-68* suppressors” to clarify these published results.